# Evaluation of Bacterial Perpetuation Assays and Plant Biomolecules Antimicrobial Activity against Cotton Blight Bacterium *Xanthomonas citri* subsp. *malvacearum*; An Alternative Source for Food Production and Protection

**DOI:** 10.3390/plants11101278

**Published:** 2022-05-10

**Authors:** Syed Atif Hasan Naqvi, Shehzad Iqbal, Umar Farooq, Muhammad Zeeshan Hassan, Muhammad Nadeem Shahid, Adnan Noor Shah, Aqleem Abbas, Iqra Mubeen, Ammara Farooq, Rehab Y. Ghareeb, Hazem M. Kalaji, Abdulwahed Fahad Alrefaei, Mohamed A. A. Ahmed

**Affiliations:** 1Department of Plant Pathology, Bahauddin Zakariya University, Multan 60800, Pakistan; atifnaqvi@bzu.edu.pk (S.A.H.N.); hafeezpp1527@gmail.com (H.-u.-R.); umarpp.uaf@gmail.com (U.F.); ranazeeshanhassan824@gmail.com (M.Z.H.); nadeemshahid7867@gmail.com (M.N.S.); 2Faculty of Agriculture Sciences, Universidad de Talca, Talca 3460000, Chile; 3Department of Agricultural Engineering, Khwaja Fareed University of Engineering and Information Technology, Rahim Yar Khan 64200, Punjab, Pakistan; ans.786@yahoo.com; 4College of Plant Science and Technology, Huazhong Agricultural University, Wuhan 430070, China; aqlpath@gmail.com; 5Key Lab of Integrated Crop Disease and Pest Management of Shandong Province, College of Plant Health and Medicine, Qingdao Agricultural University, Qingdao 266300, China; iqra.kynat1@gmail.com; 6The Institute of Molecular Biology and Biotechnology, Sargodha Campus, The University of Lahore, Sargodha 40100, Punjab, Pakistan; afc3215@gmail.com; 7Plant Protection and Biomolecular Diagnosis Department, Arid Lands Cultivation Research Institute, City of Scientific Research and Technological Applications, Borg El-Arab 21934, Alexandria, Egypt; reyassin_ghareeb@yahoo.com; 8Department of Plant Physiology, Institute of Biology, Warsaw University of Life Sciences SGGW, Nowoursynowska 159, 02-776 Warsaw, Poland; hazem@kalaji.pl; 9Institute of Technology and Life Sciences—National Research Institute, Falenty, Al. Hrabska 3, 05-090 Raszyn, Poland; 10Department of Zoology, College of Science, King Saud University, P.O. Box 2455, Riyadh 11451, Saudi Arabia; afrefaei@ksu.edu.sa; 11Plant Production Department (Horticulture–Medicinal and Aromatic Plants), Faculty of Agriculture (Saba Basha), Alexandria University, Alexandria 21531, Alexandria, Egypt; drmohamedmarey19@alexu.edu.eg

**Keywords:** *Gossypium hirsutum*, bacterium, pathology, infection, biological control, in vitro

## Abstract

Cotton (*Gossypium hirsutum*) is a global cash crop which has gained importance in earning foreign exchange for each country. Bacterial blight caused by *Xanthomonascitri* subsp. *malvacearum* (*Xcm*) has been a seriousdisease in Pakistan’s cotton belt on multiple occasions. Bacterium was isolated and identified through various biochemical and diagnostic tests wherehypersensitivity reaction, Gram staining, KOH (potassium hydroxide), catalase, starch hydrolysis, lecithinase and Tween 80 hydrolysis tests confirmed bacterium as Gram-negative and plant pathogenic. *Xcm* perpetuation assays wereevaluated on various cotton varieties under glasshouse conditions in completely randomized design by three different methods, wherein the scratch method proved to be the best upon CIM-496 and showed 83.33% disease incidence as compared with the other two methods, where Bt-3701 responded with 53.33% incidence via the spray gun method, and 50% with the water splash method on CIM-616, as compared with the control. Similarly, for disease severity percentage, Bt-3701 was pragmatic with 47.21% through scratch method, whereas, in the spray gun method, 45.51% disease severity was noted upon Bt-802, and 31.27% was calculated on Cyto-179 through the water splash method. Owing to the unique antibacterial properties of aqueous plant extracts, the poison food technique showed *Aloe vera*, *Mentha piperita*, *Syzygiumcumini* and *Azadirachta indica* with 17.77, 29.33, 18.33 and 20.22 bacterial colonies counted on nutrient agarmedium petri plate, respectively, as compared with the control. Measurement of the inhibition zone by disk diffusion technique showed *Mentha piperita*, *Syzygiumcumini*, *Citrus limon*, *Moringa oleifera* and *Syzygium aromaticum* to present the most promising results by calculating the maximum diameter of the inhibition zone, viz., 8.58, 8.55, 8.52, 8.49 and 8.41 (mm), respectively, at the highest tested concentration (75 ppm, parts per million) compared with the control. It is probable that the decoction’s interaction with the pathogen population on the host plant will need to be considered in future experiments. However, at this moment, more research into the effective management of cotton bacterial blight by plant extracts in terms of concentration determination and development of biopesticides will provide future avenues to avoid environmental pollution.

## 1. Introduction

Cotton (*Gossypium hirsutum*) is the backbone of Pakistan’s economy, accounting for more than 15% of the country’s total cultivated land [1]. It is one of the most significant commercial fiber crops in the world, providing seeds with multi-product possibilities, such as hulls, oil, lint, and animal feed [2]. This crop is attacked by various bacterial, viral and fungal diseases, and chewing and sucking insect pests, due to its extensive cultivation as a monoculture crop. These diseases, in addition to various chewing and sucking insect pests, are responsible for low cotton production around the world, resulting in heavy yield losses [3]. Black arm, angular leaf, spot of cotton and boll blight are all different names and signs of bacterial blight disease caused by *Xcm*, where bacterium has entered host plant parts through stomata, fresh wounds, natural openings and cotton bolls [4]. The characteristic symptoms of the disease appear as water-soaked lesions on both surfaces of the leaves which coalesce and later turn into angular spots, these spots are surrounded by a distinct yellow halo, and as the spots mature, the lesions expand in size and turn a black color, whereupon leaf shedding from the plant occurs [3,4,5,6]. Systemic infections appear as black streaks that resemble lightning bolts and follow the main veins of the leaf, making it difficult to distinguish between other organisms’ leaf lesions and those caused by *Xcm* [5,6]. On herbicide-damaged cotton, bacterial blight lesions are often deeper in color than lesions caused by many other cotton diseases. In advanced cases, “black arm” symptoms appear, which include dark lesions on sick leaf petioles and stems, in addition to symptoms on the bracts and bolls [7,8]. Internal boll rot can be caused by insect damage or opportunistic fungal infections, resulting in lint discoloration and seed contamination. Once fungal infections infiltrate the site and cause more boll rot, it can be difficult to tell whether a boll was originally infected by *Xcm* or another pathogen or insect [9].

The disease spreads by seed and debris in local settings and the bacteria damages the cotton plant’s vascular system directly, with the symptoms visible on the foliar sections and at all stages of plant growth [10,11,12,13,14]. The bacterium targets the cotton plant’s vascular system in the systemic phase, while the local phase causes parenchyma tissue necrosis, resulting in the loss of all foliar components, including leaves and twigs [15]. In Pakistan, during 1965, the disease was reported for the very first time in Tibba Sultan Pur, near Multan, where the Central Cotton Research Institute is situated with a 50 percent disease severity measured in most cotton fields. It was reported that the disease had the potential to diminish crop yield by up to 60% in ideal environmental conditions [16,17]. Except for a few exotic lines that are immune to all races of *Xcm*, resistant cultivars are the most cost-effective way to manage bacterial blight, however, no commercial variety has proved resistant to bacterial blight disease [18]. With the exception of resistant varieties, many management tactics have been tried in the past to deal with this problem, but none have proven effective. Although employing resistance variations to avoid this disease is a cost-effective option, there is currently no resistant high-yielding cotton cultivar available [18,19]. High-pressure sprayers, spraying on the bottom surface of the leaves, and employing carborundum as an abrasive, were once standard methods for allowing bacteria to enter plants for pathogenicity testing [20,21,22]. Plant bacterial infection is greatly aided by the application of organo silicone non-ionic wetting agents, and research on the creation of bacterial biological control agents for weeds has dramatically improved the methodology for bacterial inoculations in the field [23,24,25]. Due to the absence of resistant varieties, bacterium control is recommended using a variety of antibiotics and chemicals, either alone or in combination with other chemicals. Chemical use, on the other hand, is not recommended as ecofriendly, and eradicating the disease completely is not cost effective [25]. Many chemical control experiments have been conducted in the past, but there has been no consistent improvement in the systemic nature of the bacterium, while continued and haphazard use of chemicals leads to environmental pollution and health problems [25,26]. It is therefore, vitally important to search for alternative agents that are effective and environmentally safe for the management of pathogenic bacteria such as *Xcm* [27]. Several plant by-products containing potential biomolecules with antibacterial capabilities have been reported to be effective against a variety of pathogenic fungus and bacteria [28].Therefore, the objectives of this study were to: (1) evaluate bacterium perpetuation assays to see how different bacterial application methods affect bacterial blight development when bacterial inoculums are applied by scratch, spray gun, or water splash methods and, (2) to determine the efficacy of various local botanicals/plant extracts and their biomolecules against the bacterium.

## 2. Results

### 2.1. Bacterium Isolation, Identification, Hypersensitive and Biochemical Tests

After 24 to 48 h of incubation at 28 ± 2 °C, bacterium was isolated on nutrient agar medium with unique bright yellow colonies exuded from infected cotton leaves. The yellow raised colonies present were due to Xanthin produced by members of the genus Xanthomonads, and the colonies were small to medium, convex, and mucoid in size and shape. The injectable infiltration method was used to conduct the hypersensitivity test, which resulted in typical necrosis and necrotic symptoms on tobacco plants, developing to pustules with a raised border around the halo spot. According to the results of Gram staining, the bacterium was found Gram-negative and capable of causing biotic stress in crop plants. When the loop was removed from the slide, the KOH test corroborated the results of the Gram staining and revealed a mucoid thread. The bacterium detected bubbles and produced the catalase enzyme in the catalase test, indicating that *Xcm* protected itself from the repercussions of oxygen metabolism. The development of bubbles also suggested that the bacterium was Gram-negative. When media containing starch was stained with Lugol’s iodine after 7 days of incubation at 27 °C for starch hydrolysis, a distinct zone was observed around the bacterial growth. Similarly, the presence of dispersed zones of white and opaque colonies in culture plates showed a good response to lecithinase. After incubation at 27 °C for 7 days, the bacterium was able to hydrolyze Tween 80, revealing the presence of a dense milky white zone around the bacterial colonies, validating the genus *Xanthomonas*’ typical trait (Table 1).

### 2.2. Cotton Germplasm Response against Bacterium Perpetuation Assay under Glass House

When the pathogen was exposed to different perpetuation assays, the toothpick scratch method was found to be most effective on CIM-496, showing 83.33% disease incidence, compared with the other two methods where Bt-3701 showed 53.33% disease incidence via the spray gun method and 50% disease incidence via the water splash method on CIM-616, as compared with the control. Similarly, for disease severity percentage, Bt-3701 was pragmatic, with 47.21% achieved through the scratch method, whereas, in the spray gun method, 47.51% was noted, followed by 45.24% upon Bt-802 in the spray gun method, and Cyto-179 which was measured at 31.27% through the water splash method. The water splash method did not produce significant development of the disease. Hence, it was obvious that the scratch method was the best method to facilitate the penetration of bacterium into the infection courts (Table 2).

### 2.3. Biological Management of Cotton Blight Bacterium Xcm Using Indigenous Medicinal Plants Species/Biomolecules

The results of the antimicrobial activity of the ten different plant extracts, viz., *Syzygium aromaticum*, *Curcuma longa*, *Moringa oleifera*, *Azadirachta indica*, *Mangifera indica*, *Mentha piperita*, *Aloe vera*, *Syzygium cumini*, *Prosopis juliflora* and *Citrus limon* showed varying response against *Xcm* by both of the techniques under in vitro conditions.

### 2.4. Poison Food Technique

The results of the poison food technique showed *Mentha Piperita*, *Aloe vera*, *Azadirachta indica* and *Syzygium cumini* produced the most promising results, with 17.77, 29.33, 18.33 and 20.22 bacterial colonies counted on nutrient agar medium, compared with the control, against the infectivity of the bacterium at all tested concentrations (ppm). *Mentha Piperita* showed the best results to control the pathogen by inhibiting the bacterial colonies to 17.77 (at *p* = 0.05) at the highest concentration, followed by *Aloe vera* with 29.33 (at *p* = 0.05), and *Azadirachta indica* and *Syzygium cumini* with 18.33 and 20.22 at (at *p* = 0.05), respectively; whereas *Syzygium aromaticum*, *Curcuma longa*, *Prosopis juliflora*, *Mangifera indica*, *Syzygium cumini* and *Citrus limon* did not produce significant results (at *p =* 0.05) and failed to restrict the bacterium to one third of total colonies recorded in negative control, which was a much lower percent reduction than the control (Figure 1 and Figure 2).

### 2.5. Disk Diffusion Technique (B)

The results of the measurement of inhibition zone (MIZ) by disk diffusion technique showed *Mentha Piperita*, *Syzygium cumini*, *Citrus limon*, *Moringa oleifera* and *Syzygium aromaticum* presenting the most promising results by measuring the maximum diameter of the inhibition zone, viz., 8.58, 8.55, 8.52, 8.49 and 8.41 (mm), respectively, at the highest tested concentration (75 ppm), compared with the control. *Curcuma longa*, *Prosopis juliflora*, *Mangifera indica*, *Aloe vera* and *Azadirachta indica* did not show a significant result at the highest concentration (75 ppm), compared with the control. In this way, *Mentha piperita*, *Syzygium cumini*, *Citrus limon*, *Moringa oleifera* and *Syzygium aromaticum* resulted in 86.33, 85.66, 85.44, 85.33 and 84.44 percent reduction, respectively, over the control, when compared with the 9 cm diameter petri plate of positive control (Figure 3 and Figure 4).

## 3. Discussion

Bacterial blight, commonly known as angular leaf spot, is a bacterial disease caused by the ‘bacterium’ *Xcm* that has become increasingly prevalent in humid cotton belt regions around the world [4]. Bacterial blight infection of cotton fields can begin with contaminated crop residues from a previous season or diseased seed [4,5,6]. The pathogen survives in infested crop litter and soil, but its lifespan is uncertain and likely controlled by environmental variables. Wind-driven rain or irrigation from an infected source can spread infection (furrow or sprinkler) [4,9]. Tools, tractors, and other farm equipment can all spread *Xcm*, while bacterial blight in a field will be more severe if it emerges early in the season, particularly if plants are damaged at the seedling stage [4,7]. The bacteria can enter the plant through openings such as stomata, lenticels, and hydathodes, in addition to wounds, when plants are injured by wind-blown sand. Rainfall, especially immediately after planting, can cause rapid multiplication and spread of *Xcm* once it is established in a field [4]. Periods of heavy rainfall followed by warm and humid conditions with a relative humidity of more than 85% hasten the establishment and spread of bacterial blight once the canopy has grown [4]. Daytime temperatures of 90–100 °C and nocturnal temperatures of at least 62–68 °C are ideal for the formation and spread of bacterial blight throughout the season [4]. One *Xcm* infected seed out of 6000 is said to be enough to start a bacterial blight pandemic in a given field under ideal conditions [7,9]. Characterization and research of phenotypic diversity among *Xanthomonas* species isolated from various host plants, including cotton, has traditionally relied on non-DNA-based approaches such as physiological and biochemicals assays, and are still essential for identification of plant pathogenic bacteria to genus and species [29,30,31,32,33,34]. Accurate pathogen identification is crucial for epidemiological and ecological monitoring where several races of *Xcm* have been established, whereas a pathogen’s race is defined by its ability to trigger susceptibility or resistance in different cotton cultivars. The virulence of the bacterial blight pathogen has been reported in a number of ways by researchers from all over the world [35]. Several research workers reported their findings that no commercial varieties or lines were found resistant to *Xcm* in Pakistan [17,36]. Due to its primary infection occurring through the sowing of seed infected by *Xcm*, and the bacterium having been reported to survive for 3 to 6 months in trash applied on the surface of the soil and buried 15 cm deep, therein, commercial cotton varieties have responded differently to changing environmental conditions against bacterial blight [37,38]. Scientists have reported that the bacterium was no longer available in sterile soils after 50 to 80 days. The disease disseminates via rain splashes and air currents as a secondary mode of transmission [39]. The number of applications, plant growth stage at the time of application, plant density, and irrigation method had little or no effect in determining the susceptibility or resistance of germplasm, and the key to successful screening wasan aggressive isolate of the bacterium used for assessment purposes [40]. Spraying was only conducted early in the day when the stomata were open and bacterial infiltration was dependent on pressure spraying [40]. The authors of [41,42] discovered that the amount of infection caused by scratch, spray, or water splash inoculation was determined by the prevailing mean temperature during the infection period, rather than the temperature at the moment of inoculation. Stoughton discovered that during the infection period, a high average temperature (35 °C) encouraged disease, and that as the temperature dropped, so did the disease [42]. Temperature had an effect on the expression of resistance in partially resistant cultivars, but had no effect on immune or susceptible cultivars [43]. The authors of [44,45] also discovered that cotton leaf lesions that were 60 days old were more representative of a resistant response than cotton treated at a younger age. The top leaves were the major recipients of the bacteria in this study; therefore, while the number of leaves at the time of spraying fluctuated over time, the leaf age at treatment remained consistent. For all leaf stages treated, the lesions were the same size and had water-soaked symptoms on the underside. Experiments with *Xcm* injected directly into the main vein of leaves revealed that older leaves were more resistant than younger leaves, suggesting that the scratch method could be used in future cotton bacterial blight screening programs [46]. The novelty of current research is based mostly on the biological treatment of cotton bacterial blight, utilizing diverse plant extracts/biomolecules that are typically regarded as eco-friendly and cost-effective [46]. *Xcm* was used as the subject of the study, which was conducted in vitro. Several prior findings described various plant sections that might be used for the effective management of diseases to establish symptomless plant stands in the field under natural settings [47]. Essential oils have long been known for their antibacterial properties against pathogenic and phytopathogenic microorganisms [48,49]. *Xcm* has been killed by many essential oils from *Citrus aurantium, Citrus aurantifolia*, and *Fortunella* sp. [50]. Citral from *C. aurantifolia* inhibited *Xcm* growth, the most in disc diffusion assays, while limonene, geranyl acetate, and trans caryophllene from *Fortunella* sp. had no effect on the development of the disease [50]. Citral has a MIC of 0.5 mg/mL, indicating that substantial doses would be needed to control *Xcm* in the field. Other plant-derived chemicals may be useful in the fight against citrus canker. In vitro, water and acetone extracts from the leaf gallnuts of the Chinese “sumac” (*Rhus chinensis*) suppressed the development of *X. citri* at a concentration of 1 mg/mL [51]. When compared with the control against the bacterium’s infectivity at all concentrations tested, the results of the poison food technique revealed *Mentha Piperita, Aloe vera, Azadirachta indica*, and *Syzygium cumini*, with 17.77, 29.33, 18.33, and 20.22 bacterial colonies produced on nutrient agar, respectively. *Mentha Piperita* showed the best results to control the pathogen by inhibiting the bacterial colonies to 17.77 (*p* = 0.05) on the highest concentration, followed by *Aloe vera* with 29.33 (*p* = 0.05), *Azadirachta indica* and *Syzygium cumini* with 18.33 and 20.22 (*p* = 0.05), respectively; whereas *Syzygium aromaticum, Curcuma longa, Prosopis juliflora, Mangifera indica, Syzygium cumini* and *Citrus limon* did not produce significant results (*p* = 0.05) and failed to control the bacterium to one third of total colonies recorded in the negative control, which was a very low percent reduction over the control. Our findings were similar to the authors of [52], who used a plant extract as a biological control agent against cotton bacterial blight. The effects of *Azadirachta indica* seed oil and *Datura alba* leaf extract on the development of *Xcm* were studied at concentrations of 1, 2, and 3%. The results showed that *Datura alba* greatly controlled the bacterium’s growth at the 3% concentration. Our findings were similarly consistent with [52], who found that certain plant extracts, in addition to red and green sea weeds, were efficient in controlling *Xcm* at various concentrations. In general, *M. piperita*, which contains the active component menthol as reported by [53], had the best antibacterial action against the germs tested. Similarly, *Mentha piperita, Syzygium cumini, Citrus limon, Moringa oleifera*, and *Syzygium aromaticum*, showed the most promising results by producing the maximum diameter of the inhibition zone, viz., 8.58, 8.55, 8.52, 8.49, and 8.41 mm, respectively, at the highest tested concentration (75 ppm), as compared with the control. When compared withthe control, *Curcuma longa, Prosopis juliflora, Mangifera indica, Aloe vera*, and *Azadirachta indica* did not exhibit significant results at the highest concentration (75 ppm). When compared with the 9 cm diameter petri plate of the positive control, *Mentha Piperita, Syzygiumcumini, Citrus limon, Moringa oleifera*, and *Syzygium aromaticum* demonstrated 86.33, 85.66, 85.44, 85.33, and 84.44% reduction, respectively, over the control. The biocomposition of chemical components of plant extracts, i.e., secondary metabolites of plants, even when collected from the same species, might result in varied reactions, particularly in terms of microbe inhibition capability. Solubility, pH, volatility, diffusion characteristics in the growing medium, and the type of microbe under investigation are all elements to consider. *A. indica*, which contains the active chemical azadirachtin, was similarly found to be more effective in controlling the infection in vitro [54]. The experiment clearly showed that plant extracts have much potential against the most common cotton disease in the field, and more work will be needed in the future to treat bacterial blight-affected cotton plants with a larger number of plant extracts so that an effective bio-formulation can be created to achieve a satisfactory control of the disease [55]. The idea of using the plant’s innate resistance mechanisms to treat a wide range of diseases has generated interest in developing medications that mimic natural inducers of resistance [55]. Systemic resistance can be induced via direct activation of defense-related genes, but it can also be induced by cell priming, which results in improved elicitation of those defenses or the elicitation of additional defenses in response to pathogen attack [55]. Our findings revealed that the active chemicals in plant decoctions act directly on the pathogen and may activate resistance pathways in experimental plants, reducing disease progression. During the investigation, it was also observed that the effect of a plant decoction on the target pathogen population over time wasa factor when considering whether or not recurrent treatment would be required during the growing season [55,56]. However, more research is needed to determine the effectiveness of plant extract biomolecules for the management of bacterial blight of cotton in terms of concentration development and formulations of biopesticides in the changing environment.

## 4. Materials and Methods

### 4.1. Study Area and Planting Material

The current study was conducted at the plant pathology department, FAS and T, BZU, Pakistan. Ten cultivars, Cyto-177, 178, 179, CIM-616, CIM-496, Bt-886, Bt-986, Bt-3701, Bt-802, and SG-1, were taken from the Central Cotton Research Institute in Multan, and put in clay loam soil in earthen pots in a greenhouse.

### 4.2. Pathogen Isolation and Identification through Biochemical and Diagnostic Tests

Diseased cotton leaves with bacterial blight signs were gathered in plastic bags from the field and transferred to the lab for pathogen isolation. The diseased leaves were cleaned thoroughly with water and inoculated in petri plates containing nutrient agar. Aside from tissue bits from damaged leaves, the bacteria were isolated using a serial dilution approach, and after 24 h of inoculation, the bacterium produced yellow, circular-to-round, viscous, and mucoid colonies on nutrient agar.

### 4.3. Hypersensitivity Test, Gram Staining, KOH, Catalase, Starch Hydrolysis, Lecithinase and Tween 80 Hydrolysis

The pathogenic character of the bacterium was investigated on *Nicotiana tobaccum* plants using the injection infiltration/penetration technique, which revealed necrosis or tissue breakdown on the tobacco leaves [57]. A bacterial smear was prepared on a slide with a loop, dried and heated with a spirit lamp flame, then a dime of slide was treated for 40 s with 0.5 percent crystal violet, washed, treated with iodine solution for 30 s, decolorized with 95% ethanol, stained with 10% safranin solution for 50 s, and viewed at 100× magnification [58,59]. A 3% KOH solution was prepared to test and confirm the Gram staining results. A drop of 3% KOH solution was poured on the slide, then bacterial mass was continually stirred for 1 min by hand before pulling the loop away from the slide [60,61]. The presence of catalase enzyme in the test organism was determined using the catalase test. On a clean glass slide, a 3% hydrogen peroxide solution was applied, and a single colony was chosen and gently mixed in KOH to produce a smear in which the presence of catalase enzymes was indicated by the creation of free oxygen gas bubbles [62]. An amount of 15 g fine powdered nutrient agar was dissolved in 500 mL distilled water to create the media. Separately, 2 g rice starch was mixed with 10 mL distilled water. Continued swirling and gradual heating on a hot plate were used to incorporate the starch into the molten nutrient agar, resulting in a homogeneous mixture. Following sterilization, the medium was placed into petri plates, the bacterial culture was streaked aseptically, and the plates were incubated at 28 °C for seven days. The plates were soaked with 3% Lugol’s iodine solution after scraping the bacterial culture [63]. The egg yolk suspension was prepared from a fresh egg that had been washed with detergent under running water before being sterilized in 70% ethanol for 10 min. The egg was gently passed over a spirit lamp flame 2 to 3 times to generate a 40% *v*/*v* emulsion in distilled water, then split aseptically, and the yolk was separated in a sterile cylinder. Before pouring onto plates, 1.5 cc of a 40% *v*/*v* egg yolk emulsion was placed into molten, cooled nutritional agar substrate. Bacterial culture was injected into egg yolk media by spot and cultured for 72 h at 28 °C [64]. Media containing 5 g NaCl, 10 g peptone, 0.1 g CaCl_2_·2H_2_O and 15 g agar, was placed in 1000 mL distilled water and adjusted to pH 7.4 which was autoclaved and produced for the Tween 80 hydrolysis test, wherein 1% Tween 80 was added to the molten media and *Xac* was streaked and cultured at 27 °C for 7 d after pouring the medium into sterile petri plates. A positive or negative test was indicated by the presence or absence of opaque milky precipitate around the bacterial colonies [65].

### 4.4. Cotton Germplasm Evaluation for Bacterium Perpetuation Assay under Glass House

The pathogenicity test was performed on various cotton varieties in the glasshouse by three different perpetuation assays i.e., scratch method, which involved injuring the plants with a toothpick; spray gun method, causing injury to plant with a hand sprayer; and water splash method to observe the development of disease symptoms. Disease incidence and severity were recorded using the scale of [38], with 0 indicating no macroscopic symptoms, 1–3 indicating round dry pinhead size lesions developing (resistant), 4–6 indicating lesions turning to dry angular lesions (tolerant), 7–9 indicating lesions turning to water-soaked spots (susceptible), and 10 indicating where spots were turning to large angular water soaked lesions on leaf veins (highly susceptible). Calculations were made using the formulas below:Disease incidence *=* (number of diseased plants)/(total plants) × 100 and
Disease severity index *=* (sum of all disease rating scores)/(total number of plants) × 100/(maximum scale)

### 4.5. Biological Management of Cotton Blight Bacterium Xcm Using Indigenous Medicinal Plants Species/Biomolecules

The study of medicinal plants as a source of pharmacologically active chemicals has grown in popularity around the world, and undoubtedly plants are a potential source of antimicrobial compounds in a variety of countries, with 60 to 90% of people in underdeveloped countries using plant-derived medicine on a regular basis. Plants contain a variety of phytochemicals, such as tannins, terpenoids, alkaloids, and flavonoids, which have been found to have antimicrobial properties. While the mechanism of action and efficacy of these medicinal species is still being researched, these preparations mediate important host responses.

### 4.6. Preparation of Plant Extracts

Medicinal plant parts from ten different indigenous species were collected from the local markets of Multan and brought to the laboratory to be tested for antibacterial activity against *Xcm*, where, except for *Aloe vera*, which was removed as a gel, the dried plant parts were ground into a fine powder. In conical flasks, 15 g of ground powder was homogenized in 100 mL of autoclaved water, agitated rapidly for 15 min, and left to stand at room temperature for 6 h. All extracts were separately filtered through filter paper and centrifuged for 10 min at 8000 rpm (revolution per minutes) before being placed in a water bath for 30 min. Each plant extract was used three times at concentrations of 5, 25, 50, and 75% [39] (Table 3 and Table 4).

### 4.7. Poison Food Technique

Efficiency of each tested plant extract against the bacterium *Xcm* was assessed by calculating the number of bacterial colonies on nutrient agar media plates exclusively poisoned with specific plant extract in three replications on a colony counter (VJ-2/VJ-3; Bioevopeak). The poisoning of the petri plates was performed at four different concentrations, viz., 5, 25, 50, and 75 ppm, and the inoculated petri plates were totally randomized and incubated for 72 h at 27 °C [87].

### 4.8. Disk Diffusion Technique

Moisture was absorbed by an antibiotic-impregnated disc placed on agar previously inoculated with the test bacterium, and the antibiotic diffused outwardly through the agar medium, resulting in an antibacterial concentration gradient. Amounts of 5, 25, 50, and 75 ppm of each plant extract were impregnated for 5 to 10 s on sterilized discs of 6 mm diameter of Whatman No. 1 filter paper and placed on the surface of media on each petri plate [88].

### 4.9. Statistical Analysis

Data were computed using Microsoft Excel 2019^®^, and one-way ANOVA statistical analysis was performed using Origin 2021b. Tukey’s test was used to analyze the difference between treatment means at the 0.05 probability level. Origin 2021b was also used to conduct Sankey analysis [89].

## 5. Conclusions

The researchers determined that no cotton germplasm is completely resistant to the blight pathosystem, and that the disease attacks more quickly through fresh injuries, such as those caused by the toothpick scratch approach. Plant extracts have a strong antibacterial action against major crop plant diseases, which helps to avoid environmental pollution and toxicity. At all concentrations studied in vitro, *Mentha Piperita, Aloe vera, Azadirachta indica*, and *Syzygium cumini* were shown to be the most promising biomolecules against bacterial infection and infectivity of *Xcm*.

## Figures and Tables

**Figure 1 plants-11-01278-f001:**
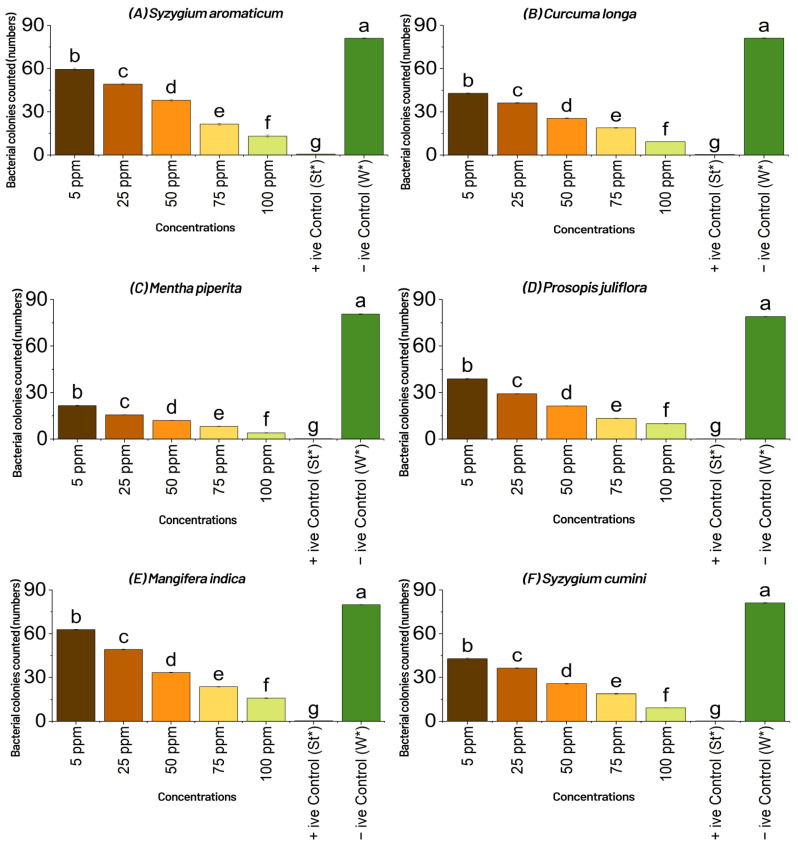
Effect of different concentrations (ppm or µg mL^−1^ = microgram per milliliter) of aqueous plant extracts, viz., (**A**)*Syzygium aromaticum*, (**B**) *Curcuma longa*, (**C**) *Moringa oleifera*, (**D**) *Azadirachta indica*, (**E**) *Mangifera indica*, (**F**) *Mentha piperita*, (**G**)*Aloe vera*, (**H**) *Syzygium cumini*, (**I**) *Prosopis juliflora* and (**J**) *Citrus limon*, on the number of bacterial colonies of *Xcm* counted on nutrient agar media by poison food technique in vitro. +ive Control (St*) = Streptomycin, which was used as standard, and similarly, –ive Control (W*) = water, which was used to compare the results with aqueous extracts (botanicals) and streptomycin. Means followed by the same letters in each column are not statistically different when compared with least significant difference (*p* < 0.05); ppm = parts per million, which can be expressed as milligrams per liter (mg/L), or the measurement of mass of a chemical, or contaminate per unit volume of water; ppm, or µg mL^−1^ = microgram per milliliter, or mg/L, have the same meaning.

**Figure 2 plants-11-01278-f002:**
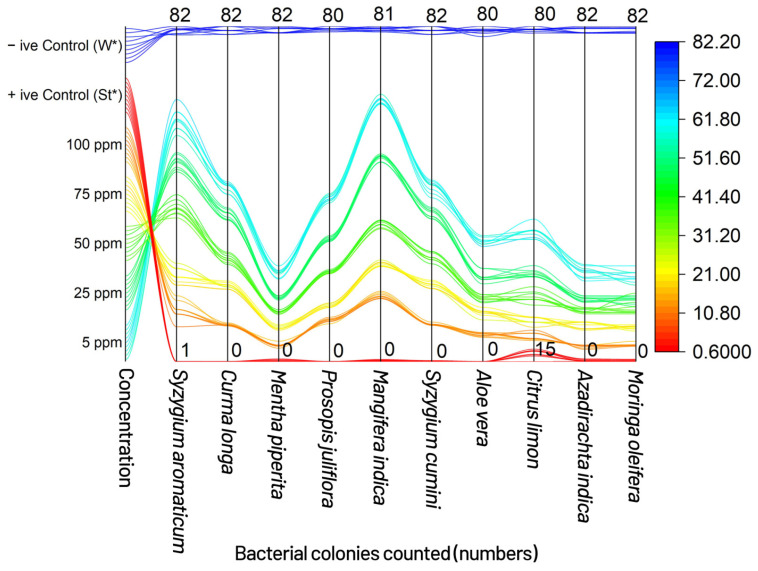
Sankey diagram showing the flow of various concentrations of treatments, communicating various levels of treatments concentrations towards counted bacterial colonies; +ive Control (St*) = Streptomycin, which was used as standard, and similarly, –ive Control (W*) = water, which was used to compare the results with aqueous extracts (botanicals) and streptomycin.

**Figure 3 plants-11-01278-f003:**
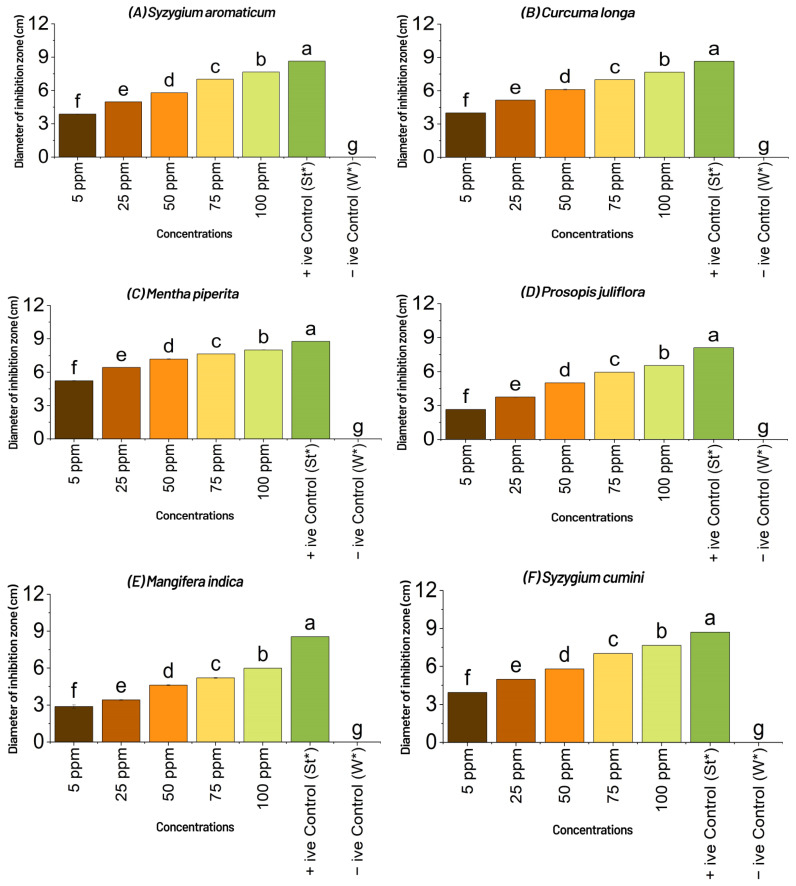
Effect of different concentrations (ppm or µg mL^−1^ = microgram per milliliter) of aqueous plant extracts, viz., (**A**)*Syzygium aromaticum*, (**B**) *Curcuma longa*, (**C**) *Moringa oleifera*, (**D**) *Azadirachta indica*, (**E**) *Mangifera indica*, (**F**) *Mentha piperita*, (**G**) *Aloe vera*, (**H**) *Syzygium cumini*, (**I**) *Prosopis juliflora* and (**J**) *Citrus limon* on the diameter of inhibition zone of *Xcm* on nutrient agar media plates by disk diffusion technique in vitro. +ive Control (St*) = Streptomycin, which was used as standard, and similarly, −ive control (W*) = water was used to compare the results with aqueous extracts (botanicals) and streptomycin. Means followed by the same letters in each column are not statistically different when compared with least significant difference (*p* < 0.05); ppm = parts per million, which can be expressed as milligrams per liter (mg/L), or the measurement of mass of a chemical, or contaminate per unit volume of water; ppm, or µg mL^−1^ = micro gram per milli liter, or mg/L, have the same meaning.

**Figure 4 plants-11-01278-f004:**
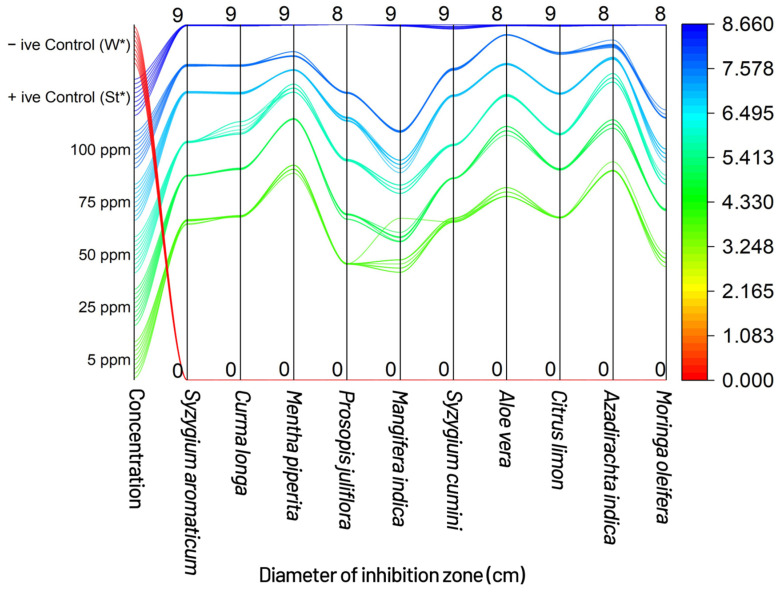
Sankey diagram showing the flow of various concentrations of treatments, communicating various levels of treatment concentrations towards the diameter of the inhibition zone (cm); +ive Control (St*) = Streptomycin, which was used as standard, and similarly, –ive Control (W*) = water, which was used to compare the results with aqueous extracts (botanicals) and streptomycin.

**Table 1 plants-11-01278-t001:** Identification of *Xcm* through biochemical and diagnostic tests on nutrient agar medium and indicator plant.

Sr. No.	Diagnostic/Biochemical Test	Reaction	Expression
1.	Hypersensitivity response	+ive	Necrotic spots
2.	Gram staining	−ive	Pink color retained
3.	KOH test	+ive	Bacterial slime showed thread
4.	Catalase test	+ive	Bubbles detected
5.	Starch hydrolysis	+ive	Zone formation
6.	Lecithinase test	+ive	Diffused zone detected
7.	Tween 80 hydrolysis	+ive	White precipitation

**Table 2 plants-11-01278-t002:** Evaluation of cotton germplasm and its response against bacterium dispersal methodology under glasshouse conditions.

Variety	Disease Incidence% (Methods)			Disease Severity% (Methods)		
	Scratch	Spray Gun	Water Splashes	Scratch	Spray Gun	Water Splashes
Cyto-177	73.33 c	53.33 b	30.00 d	40.26 bc	43.21 c	18.17 f
Cyto-178	73.33 c	53.33 b	30.00 d	45.24 e	31.91 de	26.17 c
Cyto-179	73.33 c	50.00 bc	33.33 c	32.26 b	32.24 de	31.27 b
Cim-616	50.00 d	33.33 c	50.00 b	31.34 ef	44.51 b	21.11 e
Cim-496	83.33 b	33.33 c	16.66 e	41.21 cd	36.33 dc	12.12 gh
Bt-886	50.00 d	30.00 cd	16.66 e	44.41 ef	27.21 f	17.22 fg
Bt-986	70.00 cd	50.00 bc	16.66 e	29.33 h	21.91 g	24.21 d
Bt-3701	70.00 cd	53.33 b	30.00 d	47.21 c	39.24 d	25.24 cd
Bt-802	70.00 cd	50.00 bc	30.00 d	43.64 fg	45.24 bc	19.23 f
SG-1	76.67 bc	33.33 c	33.33 c	32.68 fg	21.91 g	11.11 gh
Control	93.33 a	80.66 a	76.66 a	40.17 a	61.71 a	67.23 a

Means with the same letter are not statistically different at *p* = 0.05.

**Table 3 plants-11-01278-t003:** List of indigenous plants species/biomolecules collected to evaluate against the bacterium.

Treatments	Scientific Name	Family	Common Name	Part Used	Form Used
T1	*Syzygiumaromaticum*	Myrtaceae	Clove	Dried fruit	Fine particles
T2	*Curcuma longa*	Zingiberaceae	Haldii	Dried fruit	Fine particles
T3	*Mentha Piperita*	Labiatae	Podeena	Green leaves	Fine particles
T4	*Prosopis juliflora*	Fabaceae	Keekaar	Green leaves	Fine particles
T5	*Mangifera indica*	Anacardiaceae	Amm	Dried seed	Fine particles
T6	*Syzygium cumini*	Myrtaceae	Jamun	Dried seed	Fine particles
T7	*Aloe vera*	Xanthorrhoeaceae	Gavaargandal	Green leaves	Gel-like material
T8	*Citrus limon*	Rutaceae	Lemon	Dried fruit	Fine particles
T9	*Azadirachta indica*	Meliaceae	Nem	Green leaves	Fine particles
T10	*Moringa oleifera*	Moringaceae	Sohangana	Green leaves	Fine particles

**Table 4 plants-11-01278-t004:** Activity of plant biomolecules, active ingredients, their potential and target pathogens/diseases.

Biomolecule Source	Active Ingredient	Target Pathogen	Potential	Disease	Reference
*Syzygium aromaticum*	Sesquiterpenes, monoterpenes, hydrocarbon, phenolic compounds (e.g., Eugenyl acetate, eugenol, and β-caryophyllene	*X. oryzae* pv. *oryzae*; *Alternaria alternata*, *Fusarium chlamydosporum*, *Helminthosporumoryzae* and *Rhizoctonia bataticola*	Antiviral, antifungal, antibacterial, antioxidant	BLB of rice, Leaf spot, Blast, Black scurf disease	[66,67]
*Curcuma longa*	Curcumin, demethoxycurcumin, and bisdemethoxycurcumin	*Pythium aphanidermatum*, and *R. solani*	antifungal, antibacterial	Turmeric rhizome rot and leaf blight diseases	[68,69,70]
*Mentha Piperita*	Menthol, menthone, menthyl acetate, 1,8-cineole, limonene, beta-pinene and beta-caryophyllene	*Botrytis cinerea*, *Monilinia fructicola*, *Penicillium expansum* and *Aspergillus niger*	antifungal, antibacterial, antioxidant	Postharvest fungal disease of vegetables and fruits	[71,72,73]
*Prosopis juliflora*	C-glycosyl flavones (such as schaftoside, isoschaftoside, vicenin II, vitexin and isovitexin	*Phaeoisariopsis personata*, *Puccinia arachidis*	antifungal, antibacterial, antioxidant	Late leaf spot and rust of groundnut	[74,75]
*Mangifera indica*	lysine, leucine, cysteine, valine, arginine, phenylalanine, and methionine	*Xanthomonas* spp. *Pesudomonas* spp.	Anti-oxidant, anti-viral, immunomodulation, hypolipidemic, anti-microbial	BLB of rice, bacterial diseases	[76,77]
*Syzygium cumini*	anthocyanins, glucoside, ellagic acid, isoquercetin, kaemferol, jambosine and myrecetin	*Ascochyta rabiei*, *Xanthomnas* spp.	Antifungal, Antibacterial,	Blight disease of the chickpea	[78,79]
*Aloe vera*	sugars, lignin, saponins, salicylic acids and amino acids, Anthraquinones	*A. alternata*, *A. citri* and *A. tenuissima*	Antifungal, anti-oxidant	Leaf spots of fungi	[80,81]
*Citrus limon*	arachidonic acid, behenic acid and linoleic acid, and also tocopherols and carotenoids	*Colletotrichum gloeosporioides*	Antifungal, anti-oxidant	Anthracnose of mango fruit	[82]
*Azadirachta indica*	Azadirachtin, nimbolinin, nimbin, nimbidin, nimbidol, sodium nimbinate, gedunin, salannin, and quercetin	*Pythium aphanidermatum*, *A.alternata*, *Bipolaris sorokiniana*, *F. oxysporium*,	Antifungal, anti-oxidant	Leaf spot diseases, wilt diseases,	[83,84]
*Moringa oleifera*	myrecytin, quercetin and kaempferol	*F. oxysporum*, *F. solani* and *A. solani*, *A. alternate*	Antifungal, anti-oxidant	Wilt disease, leaf spot disease	[85,86]

## Data Availability

Not applicable.

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
