# Peer review of "Evaluation of Bacterial Perpetuation Assays and Plant Biomolecules Antimicrobial Activity against Cotton Blight Bacterium Xanthomonas citri subsp. malvacearum; An Alternative Source for Food Production and Protection"

_plants, 2022, doi:10.3390/plants11101278_

Round 1

Reviewer 1 Report

Review comments to the author

Title: ''Evaluation of bacterial dispersal methods and antimicrobial activity of indigenous medicinal plants species against Cotton Blight bacterium Xanthomonas citri subsp. malvacearum''.

Manuscript ID: plants-1719216.

Abstract:

1- Page 1, Line 42: The type of plant extracts should be mentioned.

Introduction:

1- Page 2, Line 70: The citations [3, 4, 5, 6] should be typed as [3-6].

2- Page 2, Line 82: The citations [10, 11, 12, 13, 14] should be typed as [10-14].

3- Page 2, Line 88: In the citation [16,17], add one space between the numbers.

4- Page 2, Line 97: The citations [20, 21, 22] should be typed as [20-22]. Please, apply this concept for all coming citations.

  1. Results

2.1. Bacterium isolation, identification, hypersensitive and biochemical tests

1- Page 3, Line 139: The tables were mentioned in the text starting from Table No. 2 and not No. 1, the tables must be rearranged in the text.

2.3. Biological management of cotton blight bacterium X. citri subsp. malvacearum using indigenous medicinal plants species

1- Page 4, Lines 160-161: The plant names ''Syzygium aromaticum, Curcuma longa, Moringa oleifera, Azadirachta indica, Mangifera indica, Mentha Piperita, Aloe vera, Syzygium cumini, Prosopis juliflora and Citrus limon'' should be typed in italic fonts.

2- Page 4, Line 163: The word ''in vitro'' should be typed in italic font.

2.4. Poison food technique (A)

1- Page 4, Lines 165-175: The plant names should be typed in italic fonts.

Figure 2, 4

1- The plant names should be typed in italic fonts.

  1. Discussion

1- Page 11, Line 243: The citations [43, 44, 45, 46, 47, 48] should be typed as [43-48].

1- Page 11, Line 262: The section ''[54,55] discovered that the amount...'' should be modified to '' ....et al (year)  discovered that the amount... [54, 55]''

2- Page 11, Line 268: The section ''[57,58] also discovered that 268 cotton...'' should be modified to '' ....et al (year)  also discovered that  cotton... [57, 58]''

  1. Materials and Methods

4.6. Preparation of plant extracts

1- Page 14, Line 432: Re-check the table number (Table 3 or Table 1 according to the text: Line 431)

  1. Conclusion

1- Conclusion should be supported by the results.

Abbreviations:

- List of abbreviations should be inserted by the end of the manuscript before references.

Author Response

Point to Point response file for Reviewer-I comments and suggestions

Title: ''Evaluation of bacterial dispersal methods and antimicrobial activity of indigenous medicinal plants species against Cotton Blight bacterium Xanthomonas citri subsp. malvacearum''. Manuscript ID: plants-1719216.

Dear Reviewer, Thank you very much for the valuable comments as they have helped a lot in improving the quality of the manuscript. 

Abstract:

1- Page 1, Line 42: The type of plant extracts should be mentioned.

Response: Thank you very much for your suggestion, type of plant extract has been added in the abstract at line 42

Introduction:

1- Page 2, Line 70: The citations [3, 4, 5, 6] should be typed as [3-6].

Response: Thank you very much for your suggestion, correction in the pattern of citation has been made at page 2 line 70 and showed in track changes  

2- Page 2, Line 82: The citations [10, 11, 12, 13, 14] should be typed as [10-14].

Response: Thank you very much for your suggestion, correction in the pattern of citation has been made at page 2 line 82 and showed in track changes  

3- Page 2, Line 88: In the citation [16,17], add one space between the numbers.

Response: Thank you very much for your suggestion, correction in the pattern of citation has been made at page 2 line 88 and showed in track changes  

4- Page 2, Line 97: The citations [20, 21, 22] should be typed as [20-22]. Please, apply this concept for all coming citations.

Response: Thank you very much for your suggestion, correction in the pattern of citation has been made at page 2 line 97 and the same has been implemented in whole manuscript  

  1. Results

2.1. Bacterium isolation, identification, hypersensitive and biochemical tests

1- Page 3, Line 139: The tables were mentioned in the text starting from Table No. 2 and not No. 1, the tables must be rearranged in the text.

Response: Thank you very much for your suggestion, Table numbering has been corrected in the complete manuscript and showed in track changes  

 2.3. Biological management of cotton blight bacterium X. citri subsp. malvacearum using indigenous medicinal plants species

1- Page 4, Lines 160-161: The plant names ''Syzygium aromaticum, Curcuma longa, Moringa oleifera, Azadirachta indica, Mangifera indica, Mentha Piperita, Aloe vera, Syzygium cumini, Prosopis juliflora and Citrus limon'' should be typed in italic fonts.

Response: Thank you very much for your suggestion, correction in the scientific names has been done as italicised at page 4 line 160-161 and showed in track changes  

2- Page 4, Line 163: The word ''in vitro'' should be typed in italic font.

Response: Thank you very much for your valuable comment, correction has been made at page 4 line 163 and showed in track changes  

 2.4. Poison food technique (A)

1- Page 4, Lines 165-175: The plant names should be typed in italic fonts.

Response: Thank you very much for your valuable comment, plants name has been corrected and written in italic form  

Figure 2, 4

1- The plant names should be typed in italic fonts.

Response: Thank you very much for your comment, both figures 2 and 4 plant names are now mentioned in italic form  

  1. Discussion
  • Page 11, Line 243: The citations [43, 44, 45, 46, 47, 48] should be typed as [43-48].

Response: Thank you very much for your suggestion, correction in the pattern of citation has been made at page 11 line 243 and showed in track changes 

1- Page 11, Line 262: The section ''[54,55] discovered that the amount...'' should be modified to '' ....et al (year)  discovered that the amount... [54, 55]''

Response: Thank you very much for your suggestion, correction in the pattern of citation has been made at page 11 line 262 and showed in track changes  

  • Page 11, Line 268: The section ''[57,58] also discovered that 268 cotton...'' should be modified to '' ....et al (year)  also discovered that  cotton... [57, 58]''
  •  

Response: Thank you very much for your suggestion, correction in the pattern of citation has been made at page 2 line 70 and showed in track changes 

  1. Materials and Methods

4.6. Preparation of plant extracts

1- Page 14, Line 432: Re-check the table number (Table 3 or Table 1 according to the text: Line 431)

Response: Thank you very much for your suggestion, correction in the table number has been done  

  1. Conclusion
  • Conclusion should be supported by the results.

Response: Thank you very much for your suggestion, correction has been done in the conclusion section 

Abbreviations:

- List of abbreviations should be inserted by the end of the manuscript before references.

Response: Thank you very much for your suggestion, list of abbreviations has been added in the paper before the references

Reviewer 2 Report

I have reviewed the manuscript titled: Evaluation of bacterial dispersal methods and antimicrobial activity of indigenous medicinal plants species against Cotton Blight bacterium Xanthomonas citri subsp. malvacearum.

This article aims to evaluate the efficacy of various plant extracts at various concentrations on various manifestations of bacterial blight of cotton in vitro in Pakistan's cotton. The information of this work is useful and relevant for cotton crop yield. Although, Aloe vera, Mentha piperita, Syzygium cumini and Azadirachta indica to shows potential results with 17.77, 29.33, 18.33 and 20.22 bacterial colonies counted on NA as compared to control. Measurement of inhibition zone (MIZ) by disk diffusion technique showed Mentha piperita, Syzygium cumini, Citrus limon, Moringa oleifera and Syzygium aromaticum respectively, to present most promising results by calculating the maximum diameter of the inhibition 47 zone (mm) viz., 8.58, 8.55, 8.52, 8.49 and 8.41 (mm) respectively, at the highest tested concentration (75 ppm) as compared to control. The researchers determined that no cotton germplasm is completely resistant to the blight pathosystem, and that the disease attacks more quickly through fresh injuries, such as those caused by the toothpick scratch approach in conclusion. The manuscript could be adapted by cotton growing fields especially for mitigating crop plant disease infection in the future. I think the manuscript is acceptable after major revision. The article is not innovative, however, it contains original and interesting information for cotton disease prevetntion of pretreated methods.

Abstract is well written upon and bacterium virulence was evaluated on various varieties under glass house by three different methods and scratch method proved to be the best upon CIM-496 and gave 83.33% disease incidence as compared to the other two methods. Bt-3701 responded 53.33% incidence via spray gun method and 50% with water splash method on CIM-616 as compared to control. For disease severity percentage, Bt-3701 was pragmatic with 47.21% through scratch method whereas, in spray gun method 45.51 diseases severity was noted upon Bt-802 and 31.27% was measured on Cyto-179 through water splash method.

Introduction is well addressed including Pakistan's cotton economy and total cultivated land. The information of various bacterial, viral and fungal diseases and chewing and sucking insects for low cotton production was cited. Systemic infections appear as black streaks that resemble lightning bolts and follow the main veins of the leaf, making it difficult to distinguish between other organisms' leaf lesions and those caused by Xcm. Many management measures have been tried in the past to battle this sickness, but none of them have proven to be effective with the exception of resistant varieties. Although employing resistance variations to avoid this disease is a cost-effective option, there is currently no resistant high-yielding cotton cultivar available but search for alternative agents that are effective and environmentally safe for the treatment of pathogenic bacteria information such as Xanthomonas citri subsp. malvacearum and how these plant by-products with antibacterial capabilities to be effective prevent a variety of pathogenic fungus and bacteria is important.

Materials and methods were well described.

This article would be improved if the authors revised the minor problem such as CaCl2.2H2O in line 389 at page 13.

I am not a native English speaker. The manuscript seems have no major mistakes are detected and the manuscript can be understood except some references are not well cited by using reference number and they should add author’ last name such as the problems in lines 261 and 268 at page 11, lines 302, 307 and 341at page 12 but just mentioned the reference numbers. The results of hypersensitivity response, gram staining, KOH, catalase, starch hydrolysis, lecithinase and tween-80 hydrolysis tests and plant pathogenic are well discussed. Reference fomat should be improved due to some reference use full name. It should revise to abbreviated journal name as attached file.

I enjoyed reading this manuscript; the needs of special groups of cotton growth. This manuscript presents some interesting data.

Date of this review

28 April 2022 18:58

Author Response

Point to Point response file for Reviewer-II_comments and suggestions

I have reviewed the manuscript titled: Evaluation of bacterial dispersal methods and antimicrobial activity of indigenous medicinal plants species against Cotton Blight bacterium Xanthomonas citri subsp. malvacearum.

This article aims to evaluate the efficacy of various plant extracts at various concentrations on various manifestations of bacterial blight of cotton in vitro in Pakistan's cotton. The information of this work is useful and relevant for cotton crop yield. Although, Aloe veraMentha piperitaSyzygium cumini and Azadirachta indica to shows potential results with 17.77, 29.33, 18.33 and 20.22 bacterial colonies counted on NA as compared to control. Measurement of inhibition zone (MIZ) by disk diffusion technique showed Mentha piperitaSyzygium cuminiCitrus limonMoringa oleifera and Syzygium aromaticum respectively, to present most promising results by calculating the maximum diameter of the inhibition 47 zone (mm) viz., 8.58, 8.55, 8.52, 8.49 and 8.41 (mm) respectively, at the highest tested concentration (75 ppm) as compared to control. The researchers determined that no cotton germplasm is completely resistant to the blight pathosystem, and that the disease attacks more quickly through fresh injuries, such as those caused by the toothpick scratch approach in conclusion. The manuscript could be adapted by cotton growing fields especially for mitigating crop plant disease infection in the future. I think the manuscript is acceptable after major revision. The article is not innovative, however, it contains original and interesting information for cotton disease prevetntion of pretreated methods.

Dear Reviewer, Thank you very much for the valuable comments on our manuscript as they have helped a lot in improving the quality of the manuscript. 

Abstract is well written upon and bacterium virulence was evaluated on various varieties under glass house by three different methods and scratch method proved to be the best upon CIM-496 and gave 83.33% disease incidence as compared to the other two methods. Bt-3701 responded 53.33% incidence via spray gun method and 50% with water splash method on CIM-616 as compared to control. For disease severity percentage, Bt-3701 was pragmatic with 47.21% through scratch method whereas, in spray gun method 45.51 diseases severity was noted upon Bt-802 and 31.27% was measured on Cyto-179 through water splash method.

Dear Reviewer, Thank you very much for the valuable comments upon our research conducted and their analysis, 

Introduction is well addressed including Pakistan's cotton economy and total cultivated land. The information of various bacterial, viral and fungal diseases and chewing and sucking insects for low cotton production was cited. Systemic infections appear as black streaks that resemble lightning bolts and follow the main veins of the leaf, making it difficult to distinguish between other organisms' leaf lesions and those caused by Xcm. Many management measures have been tried in the past to battle this sickness, but none of them have proven to be effective with the exception of resistant varieties. Although employing resistance variations to avoid this disease is a cost-effective option, there is currently no resistant high-yielding cotton cultivar available but search for alternative agents that are effective and environmentally safe for the treatment of pathogenic bacteria information such as Xanthomonas citri subsp. malvacearum and how these plant by-products with antibacterial capabilities to be effective prevent a variety of pathogenic fungus and bacteria is important.

Dear Reviewer, Thank you very much for the valuable comments on introduction sections and I have carefully gone through all the section to improve all quality of the manuscript. 

Materials and methods were well described.

This article would be improved if the authors revised the minor problem such as CaCl2.2H2O in line 389 at page 13.

Dear Reviewer, Thank you very much for the valuable comment, we have rectified the error pointed out by your goodself and incorporations done at page 13 line 389.  

I am not a native English speaker. The manuscript seems have no major mistakes are detected and the manuscript can be understood except some references are not well cited by using reference number and they should add author’ last name such as the problems in lines 261 and 268 at page 11, lines 302, 307 and 341at page 12 but just mentioned the reference numbers. The results of hypersensitivity response, gram staining, KOH, catalase, starch hydrolysis, lecithinase and tween-80 hydrolysis tests and plant pathogenic are well discussed. Reference fomat should be improved due to some reference use full name. It should revise to abbreviated journal name as attached file.

Dear Reviewer, Thank you very much for the valuable comments, citations issues are resolved in the complete manuscript and reference section has also been improved by observing Plants formatting. 

I enjoyed reading this manuscript; the needs of special groups of cotton growth. This manuscript presents some interesting data.

Thanks,

Regarsd!
